# Efficient Algorithms for Device Placement of DNN Graph Operators

**Jakub Tarnawski**
Microsoft Research

**Amar Phanishayee**
Microsoft Research

**Nikhil Devanur**[*]
Amazon

**Divya Mahajan**
Microsoft

**Fanny Nina Paravecino**
Microsoft

## Abstract

Modern machine learning workloads use large models, with complex structures, that are very expensive to execute. The devices that execute complex models are becoming increasingly heterogeneous as we see a flourishing of domain-specific accelerators being offered as hardware accelerators in addition to CPUs. These trends necessitate distributing the workload across multiple devices. Recent work has shown that significant gains can be obtained with *model parallelism*, i.e, partitioning a neural network's computational graph onto multiple devices. In particular, this form of parallelism assumes a *pipeline* of devices, which is fed a stream of samples and yields high throughput for training and inference of DNNs. However, for such settings (large models and multiple heterogeneous devices), we require automated algorithms and toolchains that can partition the ML workload across devices. In this paper, we identify and isolate the *structured optimization problem* at the core of device placement of DNN operators, for both inference and training, especially in modern pipelined settings. We then provide algorithms that solve this problem to optimality. We demonstrate the applicability and efficiency of our approaches using several contemporary DNN computation graphs.

## 1   Introduction

Deep Neural Networks (DNNs) have been effective across a range of applications, including image classification [KSH12, SZ14, HZRS15a], translation [WSC+16], language modeling [MKS17], and video captioning [VRD+15]. The proliferation of heterogeneous hardware accelerators [JYP+17, SPM+16] coupled with the dramatic growth in the size and the structural complexity of DNNs has bolstered the importance of *model parallelism*, where for both inference and training, the model is distributed across devices.

**DNN inference** in the "single-stream" setting [mlp], where only one inference request is issued at a time, is *latency-sensitive*. To achieve low latency, model parallel executions split the model across many accelerators [CKES16, FOP+18, CFO+18]. Model-parallel inference is beneficial due to three primary reasons. First, such splits are mandated by the memory-capacity (size) limitations of accelerators that cannot fit a single DNN model. Current DNN models have billions of parameters and require multiple GBs of space to store the weights and intermediate activations. Second, wide branching in recent DNN structures, as well as in the operator-granularity graphs for established DNNs, opens up the potential of executing data-independent sections of the computation in parallel to reduce latency. Third, the model needs to be split across multiple types of devices when a subset of operators in the graph are better suited or only supported to execute on certain accelerators.

---

[*]Work done while at Microsoft Research.

**DNN training**, on the other hand, is *throughput-bound*, as is DNN inference for the "offline" setting where many inputs can be serviced together [mlp]. Model parallelism has been proposed for training for the very same motivational reasons listed for inference above [KSH12, Kri14]. Early influential systems such as DistBelief [DCM+12] and Project Adam [CSAK14] split models to operate on commodity CPU clusters and out of CPU caches. In such a setting, operators in a DNN model are partitioned across the available devices, with each device evaluating and performing updates only on a subset of the model's parameters for all inputs. While traditional model parallel training suffers from problems of low hardware utilization, as only a single accelerator is active at any given time, **pipelined model parallelism** overcomes this deficiency. The amount of data communicated in pipelined training is the size of intermediate outputs (and corresponding gradients), which need to be sent across accelerators, and is much lower than the size of data communicated in data-parallel training. In particular, for a range of existing models that fit on a single GPU, PipeDream [HNP+18, NHP+19] uses pipelined model-parallelism to achieve much faster training time to advertised accuracy than data-parallelism. Similarly, GPipe [HCC+18, HCB+19] uses pipelined model-parallel training for very large models whose total training memory footprint exceeds the memory capacity of a single accelerator.

Given the importance of model-parallel inference and training, in this paper we present efficient algorithms to answer the following general question: *For a DNN model and a deployment scenario (a set of accelerators and their memory and interconnect constraints), how can we effectively partition the model to optimize the metric of interest, such as latency or throughput, relevant to the inference or training task at hand?*

We provide novel algorithmic approaches to tackle the problem of partitioning the model in both model-parallel inference and training scenarios, optimizing for their corresponding metrics of interest:

- Inference – (i) Model-Parallel Inference, optimized for "single-stream" latency (Figure 2a), (ii) Pipelined Inference, optimized for "offline" throughput (Figure 3a).
- Training, optimized for throughput – (i) Model-Parallel Training (Figure 2b), (ii) Pipeline-Parallel Training with PipeDream and GPipe schedules (Figure 4).

In particular, for both non-pipelined and pipelined settings, we identify the combinatorial optimization problem at the core of the device placement question, whose solution will yield the *optimal* partition. We then show how to solve this problem to optimality via Integer Programming (IP) and Dynamic Programming (DP) based algorithms. Our methods are general as they can be applied either to coarse-granularity layer graphs or to more complex fine-granularity operator graphs. We support graph partitions where accelerators can hold a *non-contiguous* fragment of the graph. We evaluate our partitioning algorithms for different scenarios described above for a variety of modern DNN workloads (7 DNNs, 16 layer and operator graphs). We find that the placements are efficient and result in non-trivial optimal splits; non-contiguous splits outperform all the techniques, with an improvement of up to $2\times$ over expert (average $1.46\times$), $2.08\times$ over local search (average $1.29\times$) [MKA07], $1.21\times$ over PipeDream (average $1.10\times$) [NHP+19], $7.69\times$ over Scotch (average $1.50\times$) [Pel09].

## 2 Related work

In the context of DNN workloads, model partitioning across different devices has mostly been a manual process driven by human experts. Most prior work on *automated* device placement falls into two broad categories. The first category comprises methods that treat the objective function (i.e., latency or throughput) as a black box. These works use heuristics, mostly based on reinforcement learning, to find partitions for a given workload (Mirhoseini et al. [MPL+17, MGP+18], Spotlight [GCL18]) or learn a placement policy that can then be adjusted for new workloads via transfer learning (Placeto [ABVG+19], GDP [ZRA+19]) or used to bootstrap a genetic algorithm (REGAL [PGN+20]). Unfortunately, these methods are computationally expensive, as they need to evaluate large numbers of placements, each of which entails a reconfiguration of the deployed devices (for a new DNN split) and measuring the runtime of several inference/training steps. For instance, [MPL+17] requires 12–27 hours of training time *on the target system* to partition modern workloads; [MGP+18] requires 12 GPU hours. For this reason, some systems (Placeto [ABVG+19], FlexFlow [JZA19]) resort to implementing a simulator to evaluate the objective.

Works in the second category – including ours – build a cost model that closely reflects real performance, and then algorithmically solve the resulting "offline" optimization problem of find-

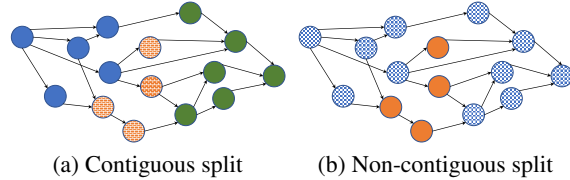

(a) Contiguous split          (b) Non-contiguous split

Figure 1: (a) Contiguous and (b) non-contiguous splits. Note that the brick-patterned orange nodes in (a) form a contiguous subgraph despite not being connected, and checked blue nodes in (b) form a non-contiguous subgraph despite being connected.

ing good partitions and schedules. This includes classic results in scheduling on multiple machines/devices [LLKS93, Gra66, KL70, PY90, SW99, ST93], as well as modern DNN scheduling works (OptCNN [JLQA18], PipeDream's [NHP+19] optimizer). Such algorithms use *profiled* compute time of each node (layer or operator) and data-transfer requirements between nodes in a graph, and the target deployment system infrastructure such as machine and network properties (e.g. measured bandwidths). Such techniques do not evaluate the performance of splits in an online fashion. Nevertheless, it has been demonstrated that for well-defined cost models the objective function closely matches real performance (PipeDream [NHP+19, Figure 15], FlexFlow [JZA19, Figure 11], OptCNN [JLQA18, Table 4]). Our throughput maximization model in Section 5 generalizes the cost model used in PipeDream [NHP+19], and our latency minimization objective (Section 4) is similar to the cost model of FlexFlow's simulator [JZA19]. In terms of approach, both OptCNN [JLQA18] and FlexFlow [JZA19] optimize over different dimensions than our methods, opting for more local parallelization strategies.

**Pipelining.** GPipe [HCB+19] and PipeDream [NHP+19] introduce *pipelined* model-parallelism for training. Given that this prior work has already shown the efficacy of pipeline parallel training on statistical efficiency (training progress compared to data-parallel training), the focus of this paper is instead on efficient algorithms to effectively partition DNN models across accelerators. For finding good DNN splits, GPipe presents no algorithm, and PipeDream proposes a method limited to layer graphs that are linear (i.e., a path). Efficiently finding optimal splits for pipelined execution in a general-DAG setting for both training and inference is the central contribution of this paper.

## 3   Computational Model

**Input.** We consider a heterogeneous system with $k$ DNN hardware accelerators and $\ell$ CPUs. For simplicity of exposition we assume all accelerators to be of the same type (such as GPU, FPGA, or TPU) for a single input. Every such accelerator has a capacity limit for its associated memory, denoted by $M$. We refer to both CPUs and accelerators as *devices*. The rest of the input to our algorithms consists of a directed acyclic graph (DAG) $G = (V, E)$ with associated weights:

- The set $V$ of nodes represents operators such as MatMul, Add, ReLu, etc. (for operator graphs), or layers such as MaxPool2d or LSTM (for layer graphs). Each node $v$ has an associated time $p_v^{\mathrm{cpu}}$ required to process $v$ on a CPU, as well as the processing time $p_v^{\mathrm{acc}}$ of $v$ on an accelerator. Each node also has a size $m_v$: the memory usage of its associated weights and activations.
- The set $E$ of directed edges encodes dependency/precedence constraints: an edge $(u, v)$ implies that the operation $v$ depends on the result of $u$. Each node $u$ has a communication cost $c_u$, which corresponds to the time required to transfer $u$'s output between CPU DRAM and the accelerator's memory, say through a PCIE bus. Crucially, this cost is paid only if $u$ and $v$ are placed on different devices: if $u$ is on an accelerator, it needs to write this output to RAM, and if $v$ is on an accelerator, it needs to read this input from RAM. We ignore the cost of reading or writing to RAM from CPUs.

**Output.** We seek to assign each node in the graph to exactly one device so that for every accelerator the sum of sizes $m_v$ of nodes assigned to it does not exceed its capacity $M$. Out of all feasible partitions we want to select one that optimizes a metric of interest (latency or throughput).

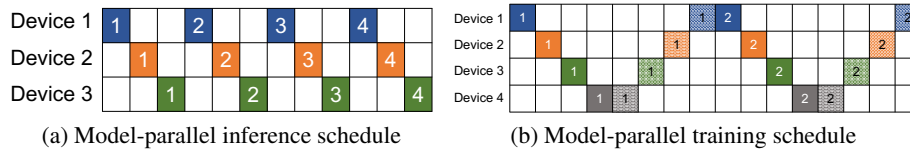

(a) Model-parallel inference schedule          (b) Model-parallel training schedule

Figure 2: (a) Single-stream model-parallel inference and (b) Model-parallel training schedule with darker shades for forward pass and lighter for backward. The $x$-axis is time, and numbers 1–4 are input minibatch identifiers. Different colors represent different devices.

**Contiguous and non-contiguous subgraphs of computation.** By default, we desire every device to hold a contiguous fragment of the DNN:

**Definition 3.1.** *We say that a set $S \subseteq V$ is* contiguous *if there do **not** exist nodes $u \in S$, $v \in V \setminus S$, and $w \in S$ such that $v$ is reachable from $u$ and $w$ is reachable from $v$.* (See Figure 1 for an example.)

This property enables subgraphs to be invoked in an uninterrupted way: all required inputs can be transferred to the accelerator at one time, after which it performs computations and produces all its outputs. This allows for simpler system implementations and less interactivity with the accelerator.

However, in this work we also explore non-contiguous splits, where the subgraphs placed on an accelerator can be arbitrary. In particular, we explain how to build a pipelined schedule for executing such a split for a stream of many samples, and how to find an optimal split of this more general form.

## 4    Inference and Latency Minimization

In this section we focus on the task of DNN inference when one sample is fed at a time (see Figure 2a). The objective here is latency, i.e., the time to produce the final output. Here, model parallelism is required and/or assists in the following two ways. First, the model might not fit in the memory of a single accelerator, making the split necessary. Second, it enables us to exploit the parallelism inherent in the model: if two operators are independent, they can be processed simultaneously if placed on different devices. We propose an *Integer Programming* based solution for this setting. For brevity, here we discuss the main ideas behind our formulation, in the simpler setting of contiguous splits. The formal Integer Programming model with detailed explanations can be found in Appendix A.

- We use binary variables $x_{vi}$ to denote whether node $v$ should be placed on device/subgraph $i$, and continuous variables $\mathrm{Latency}_v$ to denote the time at which node $v$ has finished executing and its output is available in RAM. The objective function is the maximum of $\mathrm{Latency}_v$ over all nodes $v$.
- We use variables $\mathrm{CommIn}_{ui}$ to denote the event that node $u$ produces activations that need to be transferred into subgraph $i$. We proceed similarly for outgoing data-transfers.
- For each subgraph, we have variables that control the time period during which it is processing.
- We make sure that nodes are assigned to exactly one device, and that memory size constraints are satisfied. We use a novel family of constraints to encode the contiguity requirement.
- For **non-contiguous splits**: we allow every accelerator to hold up to some number $q$ of contiguous subgraphs. We ensure that their processing times in our schedule do not overlap.
- For non-pipelined model-parallel **training** (one sample at a time, see Figure 2b), our model applies directly. A natural extra requirement is that corresponding forward and backward nodes be placed on the same device, as they operate on the same set of weights. It is easy to express this co-location constraint: for forward and corresponding backward nodes $u$ and $v$ we require $x_{ui} = x_{vi}$ for all $i$. The contiguity constraint should be enforced separately for the forward and the backward parts.

## 5    Throughput Maximization

The next goal of this work is to provide an algorithm for the setting where the DNN handles a steady stream of samples and the metric of interest is *throughput*. For simplicity we think that there are $n \to \infty$ samples to be processed offline. A schedule of choice in this scenario is model parallelism with *pipelining*. Without pipelining, only one device is active at any given time (see Figures 2a, 2b), which leads to under-utilization of resources. We remark that pipelining *schedules* that we discuss

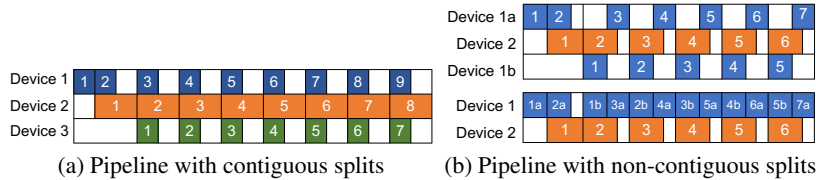

(a) Pipeline with contiguous splits      (b) Pipeline with non-contiguous splits

Figure 3: Pipelined inference. In these figures $x$-axis is time, rectangle widths are device loads (processing times for a sample), and numbers 1–9 are input sample/minibatch identifiers. The average time spent per sample is decided by the most loaded and always-busy device. In (b) the non-contiguous subgraph of device 1 can be split into two contiguous subgraphs and thought to be assigned to virtual sub-devices 1a and 1b (that can never be executing at same time). The top and bottom figures in (b) present two equivalent ways to view this schedule.

are essentially due to prior works [HCB$^+$19, NHP$^+$19], which discuss their implementation aspects, statistical efficiency, and demonstrate large real-world gains in time-to-accuracy. Here we focus on *algorithms* to find optimal *splits* for this mode of execution.

We begin by introducing our techniques in the setting of *pipelined inference*, as it is simpler yet allows us to present the main ideas. Next, we will extend them to handle *training* workloads.

## 5.1 Inference and Throughput Maximization

Imagine a DNN that has already been contiguously split into subgraphs per device. The question we ask is: How do we schedule the execution of many samples so as to maximize the throughput, or equivalently, minimize the average Time-Per-Sample? To do so, we use pipelined inference, i.e, we build a pipeline out of devices (in the order in which the subgraphs are arranged in the DNN), and we insert consecutive samples into it (see Figure 3a). Time can be viewed as divided into rounds: each sample spends one round on every device. After a short ramp-up period, the pipeline reaches a steady state, in which the duration of every round is determined by the slowest (most loaded) device. With a batch of $n$ samples, the average Time-Per-Sample becomes just the maximum load of a device (plus a vanishing $O(1/n)$ term for the ramp-up and ramp-down periods). We remark that this schedule is optimal, in the sense that that this average time cannot be lower: the bottleneck device would need to spend at least ($n \times$ its load) time to process $n$ samples in *any* schedule.

The above discussion shows that the best split is one that minimizes the maximum load of a device. Here, when searching for the best split, we do not need to simultaneously optimize for the best schedule (which was done for latency minimization using the $\text{Latency}_v$ variables in Section 4) – pipelining gives this for free. Without this scheduling aspect, we are left with a partitioning problem, which is easier to solve.

### 5.1.1 Dynamic Programming Solution

The two main ideas behind our Dynamic Programming (DP) solution are described below. First, if we want contiguous splits, then we can carve out successive device-subgraphs starting from the beginning of the network. At all times, the already-partitioned region will be a downward-closed set that we henceforth call an *ideal*.

**Definition 5.1.** *We call a set $I \subseteq V$ of nodes an* ideal *if for any $(u, v) \in E$ with $v \in I$ we have $u \in I$.*

It turns out that, going from ideal to ideal, we can obtain every possible contiguous subgraph:

**Fact 5.2.** *A set $S \subseteq V$ of nodes is contiguous (see Definition 3.1) if and only if it is the difference of two ideals: $S = I \setminus I'$ where $I' \subseteq I$.*  (The proof is given in Appendix B.1.)

General DAGs can contain exponentially many ideals (the worst case being a graph with no edges). Our second insight is that the operator graphs of most modern DNNs, while less and less linear in structure, still contain a manageable amount of branching. This topology ensures a limited number of ideals. Thus, we can consider all possible contiguous sets via Dynamic Programming.

**The Dynamic Program.** We fill a DP table of dimensions $(k + 1) \times (\ell + 1) \times$ (number of ideals in $G$), where the cell $\mathrm{dp}[I][k'][\ell']$ is intended to hold the optimal (i.e. smallest) maximum load of a device if we use $k'$ accelerators and $\ell'$ CPUs to partition the set $I \subseteq V$ of nodes. The initialization, which is $(k', \ell') = (0, 0)$, is easy: the only ideal that we can partition using 0 devices is the empty set, so we have $\mathrm{dp}[I][0][0] = 0$ if $I = \emptyset$ and $\infty$ otherwise. For $(k', \ell') \neq (0, 0)$ and any $I$, we iterate over all choices of the subgraph being placed on the last device (which is either a CPU or an accelerator), which are contiguous sets of the form $I \setminus I'$ for an ideal $I' \subseteq I$:

$$\mathrm{dp}[I][k'][\ell'] = \min_{\mathrm{ideal}\, I' \subseteq I} \min[\max\left(\mathrm{dp}[I'][k' - 1][\ell'], \mathrm{acc}(I \setminus I')\right), \max\left(\mathrm{dp}[I'][k'][\ell' - 1], \mathrm{cpu}(I \setminus I')\right)]$$

with the caveat that if $k' = 0$ or $\ell' = 0$, then we should skip the corresponding branch of the second min. By $\mathrm{cpu}(S)$ and $\mathrm{acc}(S)$ we denote the total load of the corresponding device holding the contiguous set $S$; thus $\mathrm{cpu}(S) = \sum_{v \in S} p_v^{\mathrm{cpu}}$, and $\mathrm{acc}(S)$ comprises: the incoming communication costs of $S$ ($\sum_v c_v$ over $v \notin S$ with an edge to $S$), the processing cost $\sum_{v \in S} p_v^{\mathrm{acc}}$, and the outgoing communication costs of $S$ ($\sum_v c_v$ over $v \in S$ with an edge to $V \setminus S$). If $S$ would not fit on an accelerator, i.e., $\sum_{v \in S} m_v > M$, then we instead set $\mathrm{acc}(S) = \infty$.

**Runtime and memory usage.** The DP table dominates the memory usage, which is $O(\mathcal{I} \cdot (k + 1) \cdot (\ell + 1))$, where by $\mathcal{I}$ we denote the number of ideals in $G$. It takes $O(\mathcal{I})$ time to fill one entry of the table. The entire DP solution can be implemented to run in time $O(\mathcal{I}^2 \cdot [(k+1) \cdot (\ell+1) + |V| + |E|])$, where the additional term $\mathcal{I}^2 \cdot (|V| + |E|)$ arises due to computing the costs $\mathrm{cpu}(I \setminus I')$ and $\mathrm{acc}(I \setminus I')$.

**Extensions.** A similar DP solution is used in PipeDream [NHP+19], albeit only for layer-granularity graphs that are linear (i.e., a path). That work also considers two extensions: replication (where a single subgraph is replicated on multiple devices, creating a hybrid model-parallel/data-parallel split) and hierarchical accelerator topologies (e.g. clusters of GPUs connected internally with faster interconnects). Both of these extensions can also be handled by our DP algorithm, at the costs of $O(k + \ell)$ and $O(\mathcal{I})$ factors in the runtime, respectively. See Appendix H for more details.

### 5.1.2 Dynamic Programming Solution – Linearization Heuristic (DPL)

The $\mathcal{I}^2$ term in the running time can make the DP solution inefficient for certain DNN workloads that are both large and strongly branching. To deal with this, one can reduce the search space by adding artificial edges to the graph. In particular, we use the following version of this technique: find a Hamiltonian path (in other words, a linear/topological ordering) of the input DAG using a Depth-First Search (DFS) traversal, and add this path of artificial edges. This yields the largest possible reduction of the search space: the resulting graph has only one topological ordering, and thus the number $\mathcal{I}$ of ideals becomes the number $|V|$ of nodes plus 1, giving an $O(|V|^2)$ term instead of $O(\mathcal{I}^2)$. The algorithm so obtained is polynomial-time, but it may not return the optimal solution. In Section 6 we show that it is very close to optimal for most workloads and provides a compelling trade-off between solution quality and runtime. We denote it by DPL in that section.

### 5.1.3 Integer Programming Solution

Our IP solution follows similar main ideas as that for latency minimization (Section 4). However, it is simpler as, thanks to the maximum-load objective, no scheduling aspect is present. Due to space constraints, the full formal IP formulation is given in Appendix C.

### 5.2 Non-Contiguous Splits

Suppose we are given a non-contiguous split of a DNN. We go back to the question from Section 5.1: how to best schedule our workload? Clearly, we still cannot obtain a smaller average time per sample than the max-load.[2] Fortunately, we can still match the max-load using a variant of pipelining. A challenge here is that the device-subgraphs may no longer have a linear or acyclic ordering induced from the input DNN (e.g. in Figure 1b, neither subgraph comes fully before the other). One possible

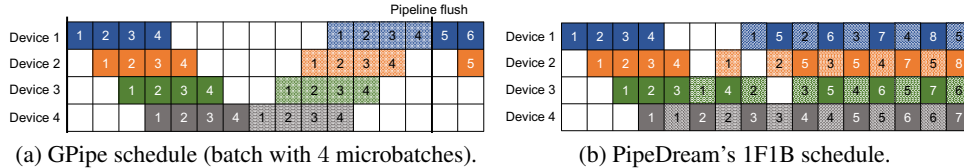

(a) GPipe schedule (batch with 4 microbatches).      (b) PipeDream's 1F1B schedule.

Figure 4: Pipeline-parallel training schedules. For simplicity the load is drawn as equal for all devices and for the forward (darker color) and backward passes (lighter color).

solution (Figure 3b for example) is to split non-contiguous subgraphs into smaller ones, so that all subgraphs can be topologically ordered, and mentally place them on *virtual devices*; then we build a pipeline of virtual devices. Now we can build a round-based schedule as before, keeping in mind that virtual devices belonging to the same real device cannot process concurrently. The bottleneck device will be the one whose total load (of all virtual devices) is maximal.[3] See Figure 3b for an example.

The above discussion shows that our max-load objective does not change when dealing with non-contiguous splits. Our IP solution (Section 5.1.3, Appendix C) "natively supports" the non-contiguous setting, by just removing the contiguity constraint.

## 5.3 Training and Throughput Maximization

Pipeline parallelism can be applied to training as well, where the task of processing large numbers of samples (and maximizing throughput) is especially relevant. As discussed at the end of Section 4, computation graphs for training consist of a forward-pass part and a backward-pass part. Certain backward nodes operate on the same state (weights/parameters) as their corresponding forward nodes, and so they must be colocated. Contiguity, if desired, should be enforced separately for the forward and the backward parts; i.e., a device $i$ would hold a contiguous subgraph of the forward part and a contiguous subgraph of the backward part. Let $\text{FW}_i$ and $\text{BW}_i$ denote their respective loads/costs.

**Objective.** GPipe [HCB$^+$19] and PipeDream [NHP$^+$19] propose two different pipeline schedules. Our max-load objective function is appropriate for both these schedules. In **PipeDream** – see Figure 4b – after a short ramp-up period, every device starts alternating between processing a forward sample and a backward sample, which together takes $\text{FW}_i + \text{BW}_i$ time. As before, the device $i$ that maximizes this quantity, i.e., the load, is the bottleneck that decides the throughput of the system. The **GPipe** schedule, shown in Figure 4a, first processes all forward samples in a batch, pipelined as they would be for inference; the average time taken for a sample in this pass is $\max_i \text{FW}_i$ (ignoring an $O(1/n)$ term). The backward pass then takes place, with an average time of $\max_i \text{BW}_i$. We thus get the objective $\max_i \text{FW}_i + \max_i \text{BW}_i$; the difference between this and the objective $\max_i \text{FW}_i + \text{BW}_i$ is insignificant, as we argue in Appendix D. For *non-contiguous splits*, both types of schedules can be modified in the same vein as in Section 5.2.

Next we describe how to extend our algorithms from Section 5.1 for training workloads. **Integer Programming:** Our IP solution handles training graphs out-of-the-box; the only required modification is that we apply the contiguity constraint (16) separately for the forward and the backward parts (if desired). **Dynamic Programming:** Our DP algorithm can only find contiguous splits, but now most devices need to be assigned two contiguous subgraphs (backward and forward). Our solution is to run the DP only on the forward part, but taking the corresponding backward nodes together with every considered contiguous subgraph (we also count their cost). See Appendix E for more details.

## 6 Experiments

In this section we evaluate our algorithms on the following modern DNN models for inference and training: BERT (with 3, 6, 12, and 24 Transformer layers), ResNet50, Inception-v3, and GNMT. Due to space constraints, we focus on the throughput (max-load) objective; the results for latency minimization can be found in Appendix F. We reiterate that we do not evaluate a particular pipelining system, but *algorithms* to find high-quality splits for pipelined executions. However, we remark that

| Workload | Nodes | DP (contiguous) | | | IP (contiguous) | | IP (non-contiguous) | | | DPL | Expert | Local search | PipeDream | Scotch |
|---|---|---|---|---|---|---|---|---|---|---|---|---|---|---|
| | | Ideals | Runtime | TPS | Runtime | TPS | Runtime | TPS | Gain | TPS | TPS | TPS | TPS | TPS |
| *Operator-granularity graphs, pipelined inference* | | | | | | | | | | | | | | |
| **BERT-3** | 235 | 1428 | **1s** | 27.92 | **1s** | 27.92 | 2s | 21.91 | 27% | 27.92 | - | 24.32 | - | 35.94 |
| **BERT-6** | 418 | 1923 | 5s | 29.58 | **4s** | 29.58 | 54s (3s*) | 28.33 | 4% | 29.58 | - | 42.52 | - | 49.80 |
| **BERT-12** | 783 | 2906 | **19s** | 147.48 | 11m (1m*) | 147.48 | >20m (18s*) | 130.03 | 13% | 147.48 | - | 257.38 | - | 230.12 |
| **ResNet50** | 604 | 241 | **0s** | 124.35 | 15s | 124.35 | 1m (10s*) | 124.35 | 0% | 124.35 | - | 250.08 | - | 197.84 |
| *Operator-granularity graphs, pipelined training* | | | | | | | | | | | | | | |
| **BERT-3** | 600 | 2774 | 8s | 65.30 | 2s | 65.30 | **1s** | 54.21 | 20% | 65.30 | - | 66.17 | - | 416.97 |
| **BERT-6** | 1071 | 3776 | 25s | 72.86 | **6s** | 72.86 | 13m (2s*) | 71.64 | 1% | 79.50 | - | 94.86 | - | 130.20 |
| **BERT-12** | 2012 | 2938 | **1m** | 438.00 | >20m (1m*) | 438.00 | >20m (1m*) | 373.42 | 17% | 438.00 | - | 737.99 | - | 800.79 |
| **ResNet50** | 1243 | 258 | **2s** | 255.19 | 2m (28s*) | 255.19 | 7s | 255.19 | 0% | 255.19 | - | 530.95 | - | 379.21 |
| *Layer-granularity graphs, pipelined inference* | | | | | | | | | | | | | | |
| **BERT-24** | 32 | 30 | **0s** | 17.79 | 1s | 17.79 | >20m (1s*) | 17.71 | 0.4% | 17.79 | 20.08 | 17.80 | 17.79 | 18.03 |
| **ResNet50** | 177 | 242 | **0s** | 33.77 | 48s | 33.77 | 14s | 33.31 | 1.3% | 33.77 | 43.92 | 35.63 | 39.38 | 34.50 |
| **InceptionV3** | 326 | 36596 | 32m | 51.55 | 3m | 51.55 | **19s** | 51.52 | 0% | 51.55 | 102.48 | 54.03 | 60.42 | 54.01 |
| **GNMT** | 96 | 17914 | 29s | 32.91 | **4s** | 32.91 | 9s | 31.68 | 4% | 32.91 | 46.21 | 31.75 | 33.03 | 34.92 |
| *Layer-granularity graphs, pipelined training* | | | | | | | | | | | | | | |
| **BERT-24** | 64 | 30 | **0s** | 41.75 | 1s | 41.75 | 9s | 39.79 | 5% | 41.75 | 49.40 | 39.93 | 41.75 | 42.01 |
| **ResNet50** | 354 | 242 | **1s** | 78.63 | 45s | 78.63 | 15s | 76.65 | 3% | 78.65 | 112.11 | 81.32 | 83.67 | 80.10 |
| **InceptionV3** | 652 | 36596 | 58m | 122.76 | 8m | 123.35 | **43s** | 117.72 | 5% | 123.93 | 213.65 | 122.80 | 128.32 | 128.32 |
| **GNMT** | 192 | 17914 | 42s | 107.00 | 4s | 107.00 | **1s** | 88.47 | 21% | 107.00 | 137.15 | 91.52 | 107.35 | 107.00 |

Table 1: Pipelined workloads for maximization of throughput / minimization of Time-Per-Sample (TPS, equal to max-load). We run the IP optimizer until it guarantees a solution within 1% of the optimum, but no longer than 20 minutes. The parenthesized times with asterisks denote the time it took the optimizer to find the solution of the final value (though it could not yet guarantee its near-optimality). DPL stands for the DP with the Linearization heuristic (see Section 5.1.2), which always runs under 3 seconds. The fastest non-DPL runtime for every input is in bold.

our max-load objective function (cost model) is a natural generalization of that of PipeDream, which has been shown [NHP+19, Figure 15] to closely reflect real performance.

**Devices and Setup.** We evaluate our algorithms on inputs corresponding to the following deployment scenarios. The DNN workloads are split across 6 accelerators of the same type (GPU for layer graphs, a hardware accelerator representing TPUs or FPGAs for operator graphs). We use 3 accelerators in case of the smaller BERT-3 and BERT-6 models. Each accelerator has 16 GB of DRAM and is connected to the CPU over a PCIE 3.0 interconnect. To assign a cost to each node and edge in the graph, we profile the workloads on GPU for layer workloads, and we estimate the numbers for the operator graphs for the hardware accelerator. More details about our experimental setup, graph topology, and implementations can be found in Appendix E. The code and workloads used for evaluations are available at `https://github.com/msr-fiddle/dnn-partitioning`.

**Baselines used for comparison.** We use the following baselines to compare our solutions:

- **Hand-crafted placements**, similar to [MGP+18, GCL18, ABVG+19]. This is still a widely used means for device placement. We perform expert splits only for layer graphs, as the operator graphs with their much stronger branching are infeasible to split manually. In line with prior work [SVL14, WSC+16], for GNMT we place each LSTM layer on a separate GPU, and then balance between 6 devices. We proceed similarly with BERT-24. In ResNet50 and Inception-v3, we split the convolution, batch normalization, and ReLu layers equally among all devices.
- **Scotch** [Pel09], a graph partitioning software used for mapping computation graphs onto devices in a balanced way, taking communication costs between dependent nodes into account. The output splits are not guaranteed to be contiguous.
- **Local search** [MKA07] is a heuristic that starts from a random split and repeatedly makes the best single-node reassignment until a local optimum is reached. We restart 10 times and take the best solution. Note that this almost always yields a non-contiguous split.
- **PipeDream** [NHP+19]'s optimizer only supports layer graphs, thus we only run it on our layer workloads. It requires the input to be a linear path, thus it contracts all branchings to single nodes.

**Results.** Table 1 shows each workload, the number of nodes (operators or layers) in the graph, runtimes of our algorithms, and the average Time-Per-Sample (TPS) – that is, the maximum device load, which is inversely proportional to throughput – of the found splits. We also report the gain

of best non-contiguous splits over best contiguous splits, and the TPS of the baselines. For better understanding of the DP runtimes we show the number of ideals in the forward part of each DNN. Visual splits of graphs obtained for the BERT-3 model by our algorithms are shown in Appendix G.

**DP vs. IP (optimality, efficiency).**    DP and IP (contiguous) both return the optimal split, so their TPS/max-load values are equal (up to a 1% IP optimality threshold). The optimization problems we solve are computationally hard, and our algorithms are exponential-time in general; however, we see that their runtimes are reasonable on real-life DNN inputs due to their workload structure. For our IP solution we used a commercial-grade solver [GO19] that ran on 4 CPU cores. It is worth noting that most of the runtime is often spent on certifying the near-optimality of the found solution; it would therefore be reasonable to cut the computation much sooner, still obtaining high-quality solutions. For the DP solution we created a single-core, self-contained implementation. Its runtimes are very competitive with those of the IP, except for the most branching models such as Inception-v3. We remark that in practice the DP runtime is dominated by the $O(\mathcal{I}^2|E|)$ term and does not depend much on the numbers $k, \ell$ of devices unless these are very large; in contrast, increasing the number of accelerators can have a large impact on the IP runtime. Moreover, the DP runtime does not depend on the node weights, whereas the IP runtime does.

**DPL (DP with the Linearization heuristic)**    (see Section 5.1.2). The DPL solution runs in time essentially $O(|V|^2|E|)$, which for our workloads is at most seconds. Crucially, the restricted search space results in a throughput loss of 9% for the BERT-6 training workload, 1% loss for InceptionV3 training, and no loss for all other workloads. Therefore, DPL would be our method of choice for very large graphs; it would be able to process graphs with tens of thousands of nodes within, say, an hour.

**Contiguous vs. non-contiguous splits.**    Our IP solution is able to find optimal non-contiguous splits. To the best of our knowledge, our work is the first one to examine *non-contiguous splits* for pipelined model parallelism; thus we use our experiments to evaluate the potential gains in throughput. We observe that on average, the best non-contiguous splits offer an ~10% gain over the best contiguous splits; for BERT-3, the gain is as large as 20-27%.

**Comparison to other baselines.**    *As seen in Table 1, our non-contiguous splits outperform all the techniques, with an improvement of up to $2\times$ over **hand-crafted expert** splits (average $1.46\times$), $2.08\times$ over **local search** (average $1.29\times$), $1.21\times$ over **PipeDream** (average $1.10\times$), $7.69\times$ over **Scotch** (average $1.50\times$).* Hand-crafted expert placements for the layer-based graphs provide 71% and 68% of the throughput in comparison to contiguous and non-contiguous splits, respectively. At the layer granularity, some workloads have a repetitive graph structure, which can be split manually across devices, yet this turns out to be not enough to obtain optimality. Furthermore, performing a reasonable human split over operator graphs is infeasible due to the large branching and number of nodes. Local search fares badly, underscoring the difficult, non-local structure of the optimization problem, which is also resistant to the heuristics used by Scotch. Finally, PipeDream only considers *linear* layer graphs and contracts branchings in the input graph; whereas our technique that does not contract branches is able to explore a larger search space for operator placement and achieve up to $1.21\times$ higher throughput.

## 7    Conclusions

In this paper we give algorithms for the problem of model partitioning of DNN workloads. They target both inference and training, and optimize the objectives of either minimizing latency or maximizing throughput. Our work follows a principled algorithmic approach, in which we identify the "right" combinatorial optimization problem to solve, and find *provably optimal* splits. While other approaches struggle to capture long-term dependencies in the graph and require trying large numbers of placements on the target system, we solve the global, end-to-end joint placement and scheduling problem in one shot. Our algorithms are efficient and can be run on arbitrary DAGs, including operator-granularity graphs, and are hardware platform agnostic. Experiments show that they outperform human experts and significantly improve over state-of-the-art methods.

## Broader Impact

This work does not present any direct foreseeable societal consequence. In general, work that makes machine learning more scalable and efficient will indirectly magnify its positive and negative impacts.

## Acknowledgments and Disclosure of Funding

This work has been funded fully by Microsoft.

## Footnotes

[2]However, the optimal max-load of a non-contiguous split can be lower than the best contiguous one.

[3]This quantity is the original load of that device, independent of the split into virtual devices.

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
