[Supplementary Material]

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

[4]Formally, $\ell$ is larger than any *antichain*: a set $A \subseteq V$ of nodes such that for any $u, v \in A$, $u$ is not reachable from $v$.

[5]Suppose we have introduced $r$ such new nodes; since each of them is free to be or not be in an ideal, the number of ideals grows by a factor $2^r$, and the DP runtime, which depends on the number of ideal pairs $I', I$ with $I' \subseteq I$, grows by a factor $3^r$.

[6]Even though the ResNet50 DNN architecture appears in both lists, these input graphs come from different sources; the layer-graph ResNet50 has runtimes profiled on a GPU, while the operator-graph ResNet50 has runtimes estimated for a non-GPU hardware accelerator. Thus the corresponding results are incomparable.

[7]We treat the CPU-related term similarly.

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

$$\min \quad \text{TotalLatency}$$

$$\text{s.t.} \quad \sum_{i=0}^{k} x_{vi} = 1 \qquad\qquad\qquad (\forall v) \qquad\qquad (1)$$

$$\text{subgraph } \{v \in V : x_{vi} = 1\} \text{ is contiguous} \qquad (\forall i = 1, ..., k) \qquad (2)$$

$$M \geq \sum_{v} m_v \cdot x_{vi} \qquad\qquad\qquad (\forall i = 1, ..., k) \qquad (3)$$

$$\text{CommIn}_{ui} \geq x_{vi} - x_{ui} \qquad\qquad (\forall(u,v) \in E)\, (\forall i = 1, ..., k) \qquad (4)$$

$$\text{CommOut}_{ui} \geq x_{ui} - x_{vi} \qquad\qquad (\forall(u,v) \in E)\, (\forall i = 1, ..., k) \qquad (5)$$

$$\text{TotalLatency} \geq \text{Latency}_v \qquad\qquad (\forall v)$$

$$\text{SubgraphStart}_i \geq \text{Latency}_v \cdot \text{CommIn}_{vi} \qquad (\forall v)\, (\forall i = 1, ..., k) \qquad (6)$$

$$\text{SubgraphFinish}_i = \text{SubgraphStart}_i + \sum_{v} \text{CommIn}_{vi} \cdot c_v$$

$$+ \sum_{v} x_{vi} \cdot p_v^{\text{acc}} + \sum_{v} \text{CommOut}_{vi} \cdot c_v \quad (\forall i = 1, ..., k) \qquad (7)$$

$$\text{Latency}_v \geq x_{v0} \cdot p_v^{\text{cpu}} \qquad\qquad (\forall v) \qquad (8)$$

$$\text{Latency}_v \geq x_{v0} \cdot p_v^{\text{cpu}} + \text{Latency}_u \qquad (\forall(u,v) \in E) \qquad (9)$$

$$\text{Latency}_v \geq x_{vi} \cdot \text{SubgraphFinish}_i \qquad (\forall v)\, (\forall i = 1, ..., k) \qquad (10)$$

$$x_{vi} \in \{0, 1\} \qquad\qquad (\forall v)\, (\forall i = 0, ..., k)$$

Figure 5: A schema of the Integer Program for latency minimization

## A  Integer Program for Latency Minimization

In this section we present our formal Integer Programming model for the problem of latency minimization.

**Computation model.** The specific way of invoking subgraphs of computation on accelerators that we assume here is motivated by production systems at a large cloud provider (anonymized), where there is no state maintained across any two subgraph invocations other than subgraph model parameters. Specifically, an accelerator, which is assigned a subgraph $S \subseteq V$ of nodes, can be *invoked* when all of its required inputs are ready in DRAM (these are outputs of nodes not in $S$ but with an edge to $S$). Once invoked, the accelerator transfers this data to its memory. Next, it processes operations $v \in S$ (in some sequential order). Finally, it transfers the results back to DRAM (these are outputs of nodes in $S$ with an edge leaving $S$). This uninterrupted mode of execution is made possible by $S$ being contiguous.

Another mild assumption we make to streamline the Integer Programming formulation is that the number $\ell$ of CPU cores is no smaller than the *width* of $G$, i.e., the maximum number of nodes that can feasibly be processed in parallel.[4]

**Our formulation.** Our IP formulation is presented in Figure 5. Devices/subgraphs of accelerators are indexed $i = 1, ..., k$, and the special index $i = 0$ denotes all CPU cores together. We use binary variables $x_{vi}$ to denote whether node $v$ should be placed on device/subgraph $i$, and continuous variables $\text{Latency}_v$ to denote the time at which node $v$ has finished executing and its output (or that of the subgraph where it is placed) is available in RAM. The objective TotalLatency is the maximum of $\text{Latency}_v$ over all nodes $v$. All variables except $x_{vi}$ are bound to be non-negative (i.e., not necessarily integral). We explain the remaining variables and constraints below:

- The variable $\text{CommIn}_{ui}$ is intended to be 1 if $u$ is not in subgraph $i$, but has an edge to it (and 0 otherwise). In this case its output needs to be transferred to the corresponding accelerator's memory. This is encoded by constraint (4).

- Similarly $\mathrm{CommOut}_{ui}$ should be 1 if $u$ is in subgraph $i$ and has an edge going out of $i$. In this case its output needs to be transferred from the corresponding accelerator to RAM. This is encoded by constraint (5).
- For a subgraph $i$, $\mathrm{SubgraphStart}_i$ is the time at which all its inputs are ready in RAM (not the accelerator's memory). This is encoded by constraint (6). $\mathrm{SubgraphFinish}_i$ is the time at which all its outputs are ready and have been transferred to RAM. Constraint (7) relates the two by taking into account the in-transfer, processing inside the subgraph, and the out-transfer.
- Constraint (1) means that every node should be assigned to exactly one subgraph (or a CPU).
- Constraint (3) encodes the requirement that the sum of sizes of nodes on accelerator $i$ should be at most $M$.
- Constraints (8) and (9) encode that node $v$ can start processing once all of its predecessors $u$ are finished. If $v$ is placed on a CPU, then its processing takes $p_v^{\mathrm{cpu}}$ time. Otherwise, its processing time is taken into account in constraint (7) of the subgraph $i$ where it is placed; the outputs of $i$ are available at time $\mathrm{SubgraphFinish}_i$, and constraint (10) will set $\mathrm{Latency}_v$ to that value.

Note that the formulation as presented in Figure 5 is not yet a Mixed-Integer Program (MIP) – but can be made so.

**Lemma A.1.** *The constraints* (2)*,* (6) *and* (10) *can be reformulated as linear constraints.*

*Proof.* To reformulate (6), take $H$ to be a very large number (guaranteed to be larger than $\mathrm{Latency}_v$ in any considered solution) and write

$$\mathrm{SubgraphStart}_i \geq \mathrm{Latency}_v - (1 - \mathrm{CommIn}_{vi}) \cdot H \,.$$

If $\mathrm{CommIn}_{vi} = 1$, then we recover the original constraint. Otherwise, if $\mathrm{CommIn}_{vi} = 0$, the right-hand side is negative and the constraint becomes vacuous. Constraint (10) can be rewritten analogously.

To formulate the contiguity constraint (2), we use extra variables $z_{vi}$, with the following linear constraints:

$$z_{vi} \geq x_{vi} \qquad\qquad (\forall v)\, (\forall i = 1, ..., k) \qquad\qquad (11)$$

$$z_{vi} \leq z_{ui} \qquad\qquad (\forall (u, v) \in E)\, (\forall i = 1, ..., k) \qquad\qquad (12)$$

$$z_{vi} \leq x_{vi} - x_{ui} + 1 \qquad\qquad (\forall (u, v) \in E)\, (\forall i = 1, ..., k) \qquad\qquad (13)$$

Intuitively, one can think of $z_{\cdot i}$ as being a non-increasing sequence that lays above $x_{\cdot i}$.

Fix $i$. We claim that the subgraph $S = \{v \in V : x_{vi} = 1\}$ is contiguous if and only if there exists a vector $(z_{vi})_{v \in V}$ satisfying constraints (11)–(13).

**"Only if" direction:** for every $v$ define $z_{vi} = 1$ if any node in $S$ is reachable from $v$, and 0 otherwise. Constraints (11) and (12) are clearly satisfied. For constraint (13), the only interesting case is when $x_{vi} = 0$ and $x_{ui} = 1$; then the constraint becomes $z_{vi} \leq 0$. This is indeed satisfied as no node $w \in S$ can be reachable from $v$; if it were, then the triple $(u, v, w)$ would contradict the contiguity of $S$ (cf. Definition 3.1).

**"If" direction:** towards a contradiction assume that there are nodes $u \in S$, $v \notin S$ and $w \in S$ such that $v$ is reachable from $u$ and $w$ is reachable from $v$. Without loss of generality assume that $(u, v) \in E$. Then $z_{vi} \leq 0$ by constraint (13). By following the path from $v$ to $w$ and repeatedly applying constraint (12) we get $z_{wi} \leq z_{vi}$, thus $z_{wi} \leq 0$. But by constraint (11) we must also have $z_{wi} \geq 1$ since $w \in S$, a contradiction. $\qquad \square$

Our formulation has $O(|V| \cdot k)$ variables and $O((|V| + |E|) \cdot k)$ constraints.

**Non-pipelined model-parallel training.** The algorithm described above can be directly applied to traditional model-parallel *training* with no pipelining (one sample at a time, as shown in Figure 2b) setting. In this case the computation graph contains a forward-pass part followed by a backward-pass part. A natural extra requirement is that corresponding forward and backward nodes be placed on the same device, as they operate on the same set of weights. It is easy to express this co-location constraint: for forward and corresponding backward nodes $u$ and $v$ we require $x_{ui} = x_{vi}$ for all $i$. The contiguity constraint (see Section A.1 below) should be enforced separately for the forward and the backward parts.

$$\min \quad \text{TotalLatency}$$

$$\text{s.t.} \quad \sum_{j=0}^{kq} x_{vj} = 1 \qquad\qquad (\forall v) \qquad\qquad (1)$$

$$\text{subgraph } \{v \in V : x_{vj} = 1\} \text{ is contiguous} \qquad (\forall j > 0) \qquad\qquad (2)$$

$$M \geq \sum_v m_v \cdot \sum_{j=(i-1)q+1}^{iq} x_{vj} \qquad (\forall i = 1, ..., k) \qquad (3^*)$$

$$\text{CommIn}_{uj} \geq x_{vj} - x_{uj} \qquad\qquad (\forall (u,v) \in E)\ (\forall j > 0) \qquad (4)$$

$$\text{CommOut}_{uj} \geq x_{uj} - x_{vj} \qquad\qquad (\forall (u,v) \in E)\ (\forall j > 0) \qquad (5)$$

$$\text{TotalLatency} \geq \text{Latency}_v \qquad\qquad (\forall v)$$

$$\text{SubgraphStart}_j \geq \text{Latency}_v \cdot \text{CommIn}_{vj} \qquad (\forall v)\ (\forall j > 0) \qquad (6)$$

$$\text{SubgraphFinish}_j = \text{SubgraphStart}_j + \sum_v \text{CommIn}_{vj} \cdot c_v$$
$$+ \sum_v x_{vj} \cdot p_v^{\text{acc}} + \sum_v \text{CommOut}_{vj} \cdot c_v \quad (\forall j > 0) \qquad (7)$$

$$\text{Latency}_v \geq x_{v0} \cdot p_v^{\text{cpu}} \qquad\qquad (\forall v) \qquad\qquad (8)$$

$$\text{Latency}_v \geq x_{v0} \cdot p_v^{\text{cpu}} + \text{Latency}_u \qquad (\forall (u,v) \in E) \qquad (9)$$

$$\text{Latency}_v \geq x_{vj} \cdot \text{SubgraphFinish}_j \qquad (\forall v)\ (\forall j > 0) \qquad (10)$$

$$\text{SubgraphStart}_j \geq \text{SubgraphFinish}_{j-1} \qquad (\forall j > 0, j \neq 1 \bmod q) \qquad (14)$$

$$x_{vj} \in \{0, 1\} \qquad\qquad (\forall v)\ (\forall j)$$

Figure 6: A schema of the Integer Program for latency minimization (non-contiguous splits: $q$ contiguous subgraphs per accelerator).

## A.1 Integer Program for Latency Minimization with Non-Contiguous Splits

Our formulation can be extended to allow every accelerator to hold up to some number $q$ of contiguous subgraphs. We then need to ensure that their processing times in our schedule do not overlap.

We use a modified Integer Program that provides for a customizable extent of non-contiguity. Here, an accelerator can be assigned several subsets of nodes $S \subseteq V$, each of which we will call a *subgraph*. The mode of computation described at the beginning of Appendix A is used for every subgraph. We require every subgraph to be a contiguous set $S$ of nodes.

We index devices/subgraphs as follows. For each accelerator $i = 1, ..., k$ we create $q$ subgraph slots indexed $j = (i-1)q+1, (i-1)q+2, ..., iq$, where $q$ is a customizable degree of non-contiguity that can be adjusted for the workload at hand. The special index $j = 0$ will denote all CPU cores together.

The modified IP formulation is given in Figure 6.

We discuss the constraints that differ from the contiguous version:

- Constraint (3*) encodes the requirement that the sum of sizes of nodes in all subgraphs that are placed on accelerator $i$ should be at most $M$.

- Constraint (14) arises because an accelerator $i$ cannot process more than one subgraph at a time. Therefore we order its subgraphs $j = (i-1)q+1, ..., iq$ by the time when they are processed.

Finally, if collocation constraints are required (e.g. for training), then they should be expressed in terms of devices rather than subgraphs. That is, for two nodes $u$ and $v$ that should be collocated, we write $x_{u0} = x_{v0}$ and for $i = 1, ..., k$, $\sum_{j=(i-1)q+1}^{iq} x_{uj} = \sum_{j=(i-1)q+1}^{iq} x_{vj}$.

Our formulation has $O(|V| \cdot q \cdot k)$ variables and $O((|V| + |E|) \cdot q \cdot k)$ constraints.

$$\text{min} \qquad \text{MaxLoad}$$

$$\text{s.t.} \qquad \sum_{i=1}^{k+\ell} x_{vi} = 1 \qquad\qquad (\forall v) \qquad (15)$$

$$\text{the subgraph } \{v \in V : x_{vi} = 1\} \text{ is contiguous \textbf{(optional)}} \qquad (\forall i) \qquad (16)$$

$$M \geq \sum_v m_v \cdot x_{vi} \qquad\qquad (\forall i = 1, ..., k)$$
$$(17)$$

$$\text{CommIn}_{ui} \geq x_{vi} - x_{ui} \qquad\qquad (\forall(u,v) \in E)\,(\forall i = 1, ..., k)$$
$$(18)$$

$$\text{CommOut}_{ui} \geq x_{ui} - x_{vi} \qquad\qquad (\forall(u,v) \in E)\,(\forall i = 1, ..., k)$$
$$(19)$$

$$\text{MaxLoad} \geq \text{Load}_i \qquad\qquad (\forall i)$$

$$\text{Load}_i = \sum_v \text{CommIn}_{vi} \cdot c_v + \sum_v x_{vi} \cdot p_v^{\text{acc}} + \sum_v \text{CommOut}_{vi} \cdot c_v \quad (\forall i = 1, ..., k)$$
$$(20)$$

$$\text{Load}_i = \sum_v x_{vi} \cdot p_v^{\text{cpu}} \qquad\qquad (\forall i = k+1, ..., k+\ell)$$
$$(21)$$

$$x_{vi} \in \{0, 1\} \qquad\qquad (\forall v)\,(\forall i)$$

Figure 7: A schema of the Integer Program for max-load minimization (throughput maximization). See Lemma A.1 on how to reformulate constraint (16) to obtain an Integer Program.

# B  Proofs

## B.1  Proof of Fact 5.2

*Proof.* **"Only if" direction:** we take $I = \{v \in V : \text{ some node in } S \text{ is reachable from } v\}$ and $I' = I \setminus S$. Clearly $I$ is an ideal and $S = I \setminus I'$; it remains to show that $I'$ is an ideal. For this, take any edge $(u, v) \in E$ with $v \in I'$; we need to show that $u \in I'$. Since $v \in I$, we also have $u \in I$. It remains to show that $u \notin S$. Assume otherwise, i.e., that $u \in S$. Since $v \in I$, some node $w \in S$ is reachable from $v$. But $v \notin S$; thus the triple $(u, v, w)$ contradicts the contiguity of $S$.

**"If" direction:** towards a contradiction assume that there are nodes $u \in S$, $v \notin S$ and $w \in S$ such that $v$ is reachable from $u$ and $w$ is reachable from $v$. We have $w \in S \subseteq I$ and $I$ is an ideal, so $v \in I$. Since $v \notin S = I \setminus I'$, we must have $v \in I'$. Since $I'$ is an ideal, also $u \in I'$. However, $u \in S = I \setminus I'$, a contradiction. $\qquad\qquad\square$

# C  Integer Program for Throughput Maximization (Max-Load Minimization)

Our IP formulation is presented in Figure 7. We index devices as follows: accelerators are assigned indices $i = 1, ..., k$ and CPUs are indexed $i = k+1, ..., k+\ell$. As in Appendix A, we use binary variables $x_{vi}$ to denote whether node $v$ should be placed on device $i$. The variables $\text{CommIn}_{vi}$, $\text{CommOut}_{vi}$ and constraints (15)–(19) are also analogous to those used in the latency-minimization IP. However, this IP is simpler as, thanks to the maximum-load objective, no scheduling aspect is present. The objective $\text{MaxLoad}$ is the maximum over $\text{Load}_i$ for all devices $i$, which is given by constraint (20) for accelerators and (21) for CPUs.

# D  Objective Functions Across Schedules

In Section 5.3 we have argued that for PipeDream schedules, the objective function that accurately reflects the quality of any split, that is, the average time taken per sample (inverse throughput), is $\max_i(\text{FW}_i + \text{BW}_i)$, where $\text{FW}_i$ and $\text{BW}_i$ are the respective loads/costs of the forward and the

Figure 8: Cumulative training time for forward and backward layers of ResNet50 (layer graph). The time accumulates with each layer progressively, that is, the $i$-th entry is the sum of processing times of layers from $1$ to $i$.

backward subgraph associated with device $i$. This is the objective function that we minimize in both our IP and DP solutions.

In the case of GPipe schedules, we have argued that the objective function can be formulated as $\max_i \mathrm{FW}_i + \max_i \mathrm{BW}_i$. This is equal to the former if the maximizing $i$'s are the same – that is, if the bottleneck device is the same for the pipelined forward pass (the first seven columns in Figure 4a) as for the pipelined backward pass (the next seven columns).

This usually holds true for real-world DNN workloads due to three factors described below:

- For any device, its forward subgraph $S$ and its backward subgraph $S'$ contain paired nodes; that is, most nodes in the backward subgraph $S'$ have a corresponding forward node, which, due to colocation constraints, will be in $S$, and vice versa. For instance, most forward nodes operate on a set of weights, for which the backward pass then computes gradients and weight updates.

- The processing and communication times of such corresponding/colocated nodes are correlated; for example, if the forward node corresponds to a matrix multiplication, then the processing times of both forward and backward nodes will grow with the size of the matrix.

- In fact, GPipe uses a re-materialization technique [CXZG16] to save memory: it discards stashed activations generated in the forward pass (needed later in the backward pass), and instead reruns the forward pass operators in the backward pass to re-materialize the required stashed activations for the backward operators. If this is reflected in the DNN workload operator-graph or layer-graph, then it further increases the aforementioned correlation between forward and backward times.

In Figure 8 we plot cumulative forward and backward times for an example training workload (that does not use re-materialization), which grow at a similar pace. These runtimes have been profiled on a GPU.

The above discussion motivates the use of our objective $\max_i(\mathrm{FW}_i + \mathrm{BW}_i)$ as a proxy for the objective $\max_i \mathrm{FW}_i + \max_i \mathrm{BW}_i$ also in the case of GPipe schedules. Nonetheless, our IP solution can also be adjusted to optimize the latter objective. Unsurprisingly, we empirically find that splits found by optimizing either objective differ by at most 6% when using re-materialization.

## E  Throughput Maximization – Implementation, Further Details and Results

The code of our implementations and the DNN workloads we used as inputs can be found at `https://github.com/msr-fiddle/dnn-partitioning`.

This section extends and provides more details for Section 6.

- In Section E.1 we discuss the computing setup used to run our experiments, how the input workloads are obtained, and more details on how our solutions (especially the Dynamic Programming method) preprocess the input before invoking the core algorithm.

| Workload | DP | IP (contiguous) | IP (non-contiguous) | Expert | Local search | PipeDream | Scotch | Single acc. |
|---|---|---|---|---|---|---|---|---|
| Operator-granularity graphs, pipelined inference | | | | | | | | |
| **BERT-3** | 1.00× | 1.00× | 1.27× | - | 1.15× | - | 0.78× | 0.57× |
| **BERT-6** | 1.00× | 1.00× | 1.04× | - | 0.70× | - | 0.59× | 0.38× |
| **BERT-12** | 1.00× | 1.00× | 1.13× | - | 0.57× | - | 0.64× | 0.23× |
| **ResNet50** | 1.00× | 1.00× | 1.00× | - | 0.50× | - | 0.63× | 0.38× |
| Operator-granularity graphs, pipelined training | | | | | | | | |
| **BERT-3** | 1.00× | 1.00× | 1.20× | - | 0.99× | - | 0.16× | 0.53× |
| **BERT-6** | 1.00× | 1.00× | 1.02× | - | 0.77× | - | 0.56× | 0.38× |
| **BERT-12** | 1.00× | 1.00× | 1.17× | - | 0.59× | - | 0.55× | 0.23× |
| **ResNet50** | 1.00× | 1.00× | 1.00× | - | 0.48× | - | 0.67× | 0.37× |
| Layer-granularity graphs, pipelined inference | | | | | | | | |
| **BERT-24** | 1.00× | 1.00× | 1.00× | 0.89× | 1.00× | 1.00× | 0.99× | 0.19× |
| **ResNet50** | 1.00× | 1.00× | 1.01× | 0.77× | 0.95× | 0.86× | 0.98× | 0.17× |
| **InceptionV3** | 1.00× | 1.00× | 1.00× | 0.50× | 0.95× | 0.85× | 0.95× | 0.17× |
| **GNMT** | 1.00× | 1.00× | 1.04× | 0.71× | 1.04× | 1.00× | 0.94× | 0.18× |
| Layer-granularity graphs, pipelined training | | | | | | | | |
| **BERT-24** | 1.00× | 1.00× | 1.05× | 0.85× | 1.05× | 1.00× | 0.99× | 0.19× |
| **ResNet50** | 1.00× | 1.00× | 1.03× | 0.70× | 0.97× | 0.94× | 0.98× | 0.17× |
| **InceptionV3** | 1.00× | 1.00× | 1.04× | 0.57× | 1.00× | 0.96× | 0.96× | 0.17× |
| **GNMT** | 1.00× | 1.00× | 1.21× | 0.78× | 1.17× | 1.00× | 1.00× | 0.21× |

Table 2: Throughput maximization results, same as in Table 1 in Section 6, but presented in terms of throughput improvement in relation to the DP (Dynamic Program, contiguous splits) being $1\times$. For example, on BERT-3 inference operator-graph, the best non-contiguous split offers $1.27\times$ the throughput of the best contiguous one, and Scotch gives $0.78\times$ the throughput of the best contiguous one. In addition, we show the single-accelerator throughput (placing the entire DNN workload on one accelerator). See Figure 9 for a graphical representation of data in this table.

- In Section E.2 we present the numerical results of the experimental evaluation in Section 6 in an equivalent form, comparing the throughput of the baselines to that of our Dynamic Programming (contiguous) algorithm.

- In Section E.3 we measure the throughput advantage that can be obtained by using finer-granularity operator graphs in lieu of layer graphs.

### E.1 Implementation and Experimental Setup

**Computing setup.** All our experiments are executed on a machine with an Intel Xeon E5-2673 v4 CPU and 64 GB of RAM running Ubuntu 18.04. The dynamic programming solution is implemented in C++ and compiled with gcc 7.4 using the -O3 optimization flag; it is a sequential (single-threaded) implementation. The Integer Programming formulations are solved using Gurobi 8.1 [GO19], which runs on 4 CPU cores. The IP models are constructed using Gurobi bindings for Python; the runtime of this construction is insignificant.

**Inputs (workloads).** We benchmark our algorithms on diverse and widely used deep learning workloads ranging from transformer models (BERT) and convolutional neural networks (ResNet, Inception) to translation LSTM-based models (GNMT). We exported BERT [VSP+17] and ResNet [HZRS15b] operator graphs through the ONNX Runtime library [ONN20]. It allows exporting the operator graph topology for deep learning models by taking as input their forward pass and appending the corresponding backward pass to generate an output in ONNX format. We obtained all the layer graphs from previous work [HNP+18]. All the graphs (both operator- and layer-granularity) have the node runtimes profiled or estimated; we then convert the topology of each graph to a JSON format, comprising all the relevant information about the graph that is required of an input instance of our algorithms (see Section 3).

**DP preprocessing.** In our Dynamic Programming solution we need to handle colocation constraints given in the input: certain pairs of nodes operate on the same state and thus they are required to be on the same device. A common scenario where this arises concerns forward and backward nodes that

Figure 9: An illustration of throughput maximization results from Table 2, with DP (contiguous) serving as $1\times$. The blue bars are algorithms from this work, whereas the non-blue-colored bars show baselines. Plots (a) and (b) represent throughput improvements for operator-level graphs, and (c) and (d) for layer-level graphs.

operate on the same set of weights, but pairs of forward nodes (or pairs of backward nodes) can also be colocated. In the input files this is expressed via the `colorClass` field: nodes of the same color class must be placed on the same device.

Moreover, for training workloads, the DP can natively find only contiguous splits, but now most devices need to be assigned two contiguous subgraphs (backward and forward). Therefore we run the DP only on the forward part, but we take the corresponding backward nodes together with every considered contiguous subgraph. However, some care is required to make sure that we assign those backward nodes that do not have a corresponding forward node; we call these backward nodes *orphaned*.

For the reasons outlined above, our solution needs to run a series of preprocessing steps before the core DP method can be applied:

- For every color class $C \subseteq V$, i.e., a set of nodes that must be colocated, let $C_{\text{FW}}$ and $C_{\text{BW}}$ be the forward and backward nodes in $C$, respectively (so that $C = C_{\text{FW}} \cup C_{\text{BW}}$). We *contract* each set $C_{\text{FW}}$ and each set $C_{\text{BW}}$ (that is, we compress each of them into a single node; this new node will be forward for $C_{\text{FW}}$ and backward for $C_{\text{BW}}$).

- The input graph was guaranteed to be acyclic at the beginning, but the new contracted graph may no longer be acyclic. For instance, there could be a path $u, v, w$ where $u$ and $w$ are colocated (but not $v$); then the contracted graph will have edges in both directions between $v$ and the new node corresponding to $\{u, w\}$. In the original graph, any colocation-respecting contiguous split would need to contain all of $u, v, w$ in a single subgraph; more generally, every strongly connected component in the contracted graph needs to be colocated. Therefore, we contract all strongly connected components. Now the contracted graph is again acyclic.

- Later, when we run the DP, while considering a subgraph $S$ of forward nodes we will consider the subgraph $S'$ of their corresponding backward nodes at the same time, and take the total computation and communication cost of $S \cup S'$ into account. Thus, when we have assigned all forward nodes, we will have also assigned all backward nodes that are not orphaned. However, orphaned nodes would not be assigned to any subgraph/device.

  To prevent this behavior, we introduce new artificial forward nodes, to be images of the orphaned backward nodes. When the DP decides where to place these new forward nodes, it will also have decided the placement of the orphaned backward nodes. (At the end we will remove the artificial nodes from the final split.)

  However, if the new nodes are isolated (have no adjacent edges), then the number of ideals grows exponentially[5]; furthermore, as the placement of the new forward and orphaned backward nodes is arbitrary, we may end up with non-contiguous splits on the backward side.

  To deal with these issues, we also add new artificial edges adjacent to the new artificial nodes. Since backward nodes and edges mostly resemble a mirror image of their corresponding forward nodes and edges, we add the new edges in such a way as to also build such a mirror image. Namely, for a backward edge $(u', v')$ where at least one of $u', v'$ is orphaned, we add a forward edge $(v, u)$, where $u$ and $v$ are the forward images of $u'$ and $v'$ respectively (note that at least one of $u, v$ is a new artificial node).

After these preprocessing steps, we can use our core DP method on the contracted graph. Once this is done, we map the resulting splits back to the original graph and return the result. For more details on implementation, see the attached code and the comments therein.

We remark that due to our preprocessing steps, the number of ideals may sometimes be smaller than the number of nodes in the initial input graph (this happens for several of our workloads in Table 1).

**Non-uniform outgoing communication costs.** In the case of operator graphs, the input files for our solvers are created based on ONNX computation graphs. There, communication costs are given on edges, rather than on nodes as we require in our model (see Section 3). In the vast majority of

| Workload | DP (run on original operator graph) | DP (run on contracted layer graph) | Gain |
|---|---|---|---|
| Operator-granularity graphs, pipelined inference | | | |
| **BERT-3** | 27.92 | 27.92 | 0% |
| **BERT-6** | 29.58 | 29.58 | 0% |
| **BERT-12** | 147.48 | 159.43 | 8% |
| **ResNet50** | 124.35 | 129.15 | 4% |
| Operator-granularity graphs, pipelined training | | | |
| **BERT-3** | 65.30 | 65.30 | 0% |
| **BERT-6** | 72.86 | 72.86 | 0% |
| **BERT-12** | 438.00 | 465.41 | 6% |
| **ResNet50** | 255.19 | 269.63 | 6% |

Table 3: Throughput maximization; throughput advantage of optimization on the operator-granularity level vs. the layer-granularity level (see Section E.3), for optimal contiguous splits.

cases, all edges going out of the same node $u$ have the same cost, and we can set that cost as parameter $c_u$. However, sometimes there could be two or more edges with different costs going out of the same node in an ONNX graph; this situation corresponds to e.g. sending different parts of the operator's output to different operators. In this case, we perform the following reduction:

Suppose that $u$ has outgoing edges to nodes $v_1, v_2, ..., v_r$ with possibly different edge costs $d_1, d_2, ..., d_r$. For every outgoing edge $(u, v_j)$, we *subdivide* it: insert a new node $w_j$ in the middle and replace the edge $(u, v_j)$ with two edges $(u, w_j)$ and $(w_j, v_j)$. The new node $w_j$ should have $p_{w_j}^{\text{cpu}} = p_{w_j}^{\text{acc}} = m_{w_j} = 0$ and be colocated with $u$. We set $c_{w_j} = d_j$. Finally, set $c_u$ to any value, say $\infty$; this communication cost will never be paid in any feasible solution, as now $u$ is colocated with all of its successors, which are $w_1, w_2, ..., w_r$.

After obtaining a final split, we may remove the artificial nodes $w_j$ from the solution. It is easy to see that the way we have set the outgoing communication costs $c$ on nodes reflects the edge-communication costs given in the input ONNX graph.

## E.2 Throughput Advantages of Our Algorithms

In this section we present the numerical results of the experimental evaluation in Section 6 in an equivalent form, displaying the throughput advantages obtained by our algorithms with respect to baselines. See Table 2 on page 19 and Figure 9 on page 20.

## E.3 Advantage of Operator vs. Layer Graphs

In this section we measure the throughput advantage that can be obtained by using finer-granularity operator graphs in lieu of layer graphs. No conclusions on this matter can be drawn from the experimental results of Section 6 or Appendix F alone, as our operator-graph and layer-graph workloads are disjoint.[6] Therefore we proceed as follows: for each operator-graph workload, we manually annotate all nodes to group them into corresponding layers. Then we contract each layer and run the DP algorithm on the layer-graph thus obtained.

We present the results of this experiment in Table 3. We compare the optimal contiguous splits. The results show that finding the best split on the more precise operator level results in a throughput advantage of up to 8%.

## F  Experiments – Latency Minimization

In this section we evaluate our Integer Programming (IP) based algorithm for latency minimization. We consider the most relevant deployment scenario: single-sample inference with memory-bound

accelerators (that is, when the entire model does not fit on one accelerator). We run our algorithm for the same inference workloads as in Section 6. As before, we use Gurobi to solve our IP formulation.

**Devices, implementation, experimental setup.**    We run experiments on the same inference workloads as in Section 6. However, to model a memory-bound deployment scenario where splits are necessary to fit the DNN, we assume an accelerator DRAM size of either 600 MB (for smaller DNNs, of size at most 3.6GB) or 2 GB (for larger DNNs, of size at least 9GB), and a number of accelerators such that the total accelerator memory is 1.4–1.8 times the size of the DNN. Note that this implies, in particular, that a single-accelerator split is not feasible for any of our workloads. In keeping with the mild assumption made at the beginning of Appendix A, we assume 8 CPU cores. For other implementation details, please refer to Appendix E.1.

**Baselines.**    We compare our IP algorithm against four baseline solutions. The **first** is the following **greedy** algorithm:

- contract colocated nodes and any strongly connected components that arise (as in Appendix E),
- fix a topological ordering of the nodes,
- for every available accelerator, place as many nodes (in the topological order above) as will fit on the accelerator,
- place any remaining nodes on the CPU.

The greedy algorithm returns a contiguous split that is feasible (i.e. satisfies the memory size constraints). For all our test workloads, it is able to place all nodes on accelerators (thus it does not use the CPU). However, it does not take processing times or communication costs into account when selecting the split. The runtimes of this baseline are under 0.5s.

Our **second** baseline is meant to answer the following question: *If we obtain splits by optimizing the max-load objective, as we would for the throughput maximization task (that pertains to the pipelined setting), are they "good" in terms of minimizing latency as well?* Therefore, we obtain contiguous splits by running the **max-load DP** algorithm of Section 5.1.1, and then we report the single-sample latency that they obtain. The runtimes of this baseline are essentially the same as those of the max-load DP reported in Section 6 for the corresponding DNNs.

The **third** baseline is **Scotch** [Pel09], a graph partitioning software used for mapping computation graphs onto devices in a balanced way, taking communication costs between dependent nodes into account (used also in Section 6). It produces non-contiguous splits.

The **fourth** baseline are **human-expert** placements, the same as used in Section 6.

We do not compare against a local search heuristic, as it is not clear how to design one that satisfies the memory bounds.

**Results.**    Table 4 shows each workload, the number of nodes (operators or layers) in the graph, and the latencies found by our IP algorithm and by the baselines. We also report running times.

As we remarked in Section 5, the latency minimization task is significantly harder than throughput maximization as it contains a scheduling component. This is reflected in the performance of our IP algorithm: for five out of eight workloads used, the IP solver did not converge to certified (near-)optimality within 1 hour. However, it still comes out far ahead:

- The IP, even where it could not prove that it has found an optimal solution, does no worse than the baselines. In fact, it outperforms the best of them by a margin of around 20% in terms of the solution value (latency) for half of the considered workloads.

- Similarly as for max-load minimization (Section 6), we note that the solution quality improves slowly over time, and most of the runtime is often spent on certifying the near-optimality of the found solution; it would therefore be reasonable to cut the computation much sooner, still obtaining high-quality solutions.

- In particular, for each workload it took the IP solver at most 7 minutes to match the solution quality of the best baseline.

| Workload | Nodes | Greedy | Max-load DP | Scotch | Expert | IP | | | |
|---|---|---|---|---|---|---|---|---|---|
| | | Latency | Latency | Latency | Latency | Latency | Runtime | MIP Gap | Gain |
| Operator-granularity graphs, single-query inference | | | | | | | | | |
| BERT-3 | 235 | 416.20 | 415.90 | 497.75 | - | **408.47** | 3m (6s*) | <1% | 1.8% |
| BERT-6 | 418 | 494.13 | 445.48 | 564.61 | - | **438.06** | >1h (1m*) | 12.6% | 1.7% |
| BERT-12 | 783 | 867.84 | 1327.03 | 1755.41 | - | **729.56** | >1h (10m*) | 93.8% | 19.0% |
| ResNet50 | 604 | 839.54 | 1123.65 | 857.73 | - | **672.06** | >1h (7m*) | 54.0% | 24.9% |
| Layer-granularity graphs, single-query inference | | | | | | | | | |
| BERT-24 | 32 | **100.22** | 108.03 | 108.03 | 111.94 | **100.22** | 14s (1s*) | <1% | 0.0% |
| ResNet50 | 354 | 4197.06 | 1443.79 | 3610.87$^\dagger$ | OOM | **1191.02** | >1h (19m*) | 93.1% | 21.2% |
| InceptionV3 | 652 | 2485.24 | 1621.74 | 3068.00$^\dagger$ | OOM | **1318.08** | >1h (43m*) | 93.5% | 23.0% |
| GNMT | 192 | 268.50 | 244.33 | 636.91$^\dagger$ | 293.40$^\dagger$ | **225.6** | 3m (1m*) | <1% | 8.3% |

Table 4: Single-sample inference workloads for latency minimization. We run the IP optimizer until it guarantees a solution within 1% of the optimum, but no longer than 60 minutes. Where the optimization was terminated after 60 minutes, we report the optimality gap that the solver was able to certify at that time. The parenthesized times with asterisks denote the time it took the optimizer to find a solution within 2% of the final value (though it could not yet guarantee its near-optimality). We also report the latencies obtained by the four baselines described in Section F; their running times are always under 0.5s (Greedy, Scotch) or the same as reported in Section 6 (Max-load DP). Daggers$^\dagger$ denote a slight (between 20% and 34%) violation of the memory constraints, and "OOM" denotes a major violation (more than a factor $3\times$). The best latency for each workload is given in bold. In the column "Gain" we report the advantage of our IP algorithm's solution over the best baseline.

Figure 10: Optimal contiguous (top) and non-contiguous (bottom) splits of the BERT-3 operator-level inference graph onto 3 accelerators and 1 CPU (for throughput maximization). Each node is colored based on its placement – red color indicates CPU placement, and each remaining color indicates a different accelerator. The non-contiguous split achieves a 27% higher throughput. If viewed on a computer, the figures can be zoomed in to an arbitrary degree for better inspection.

**Comparison to baselines.**

- **Greedy**: our algorithm achieves latency lower by up to 72% (i.e., over $3\times$ faster inference; 23% lower latency on average).

- **Max-load DP**: the latency-IP achieves lower by up to 42% (17% on average). This shows that the best splits for latency minimization are, indeed, different from the best splits for max-load minimization (throughput maximization for pipelined settings). Still, the max-load DP turns out to be the best baseline in 5 out of the 8 cases, showing some degree of compatibility between the two objectives.

- **Scotch**: our algorithm achieves latency lower by up to 67% (40% on average). In fact, Scotch never does better than the greedy heuristic. Furthermore, as it does not balance devices with respect to memory usage, it violates the memory constraints by up to 34% in some cases.

- **Human expert** splits: as in Section 6, we provide them for layer graphs only, due to the large node counts and high branching of operator graphs. As the expert splits were not designed with our strictly memory-bound scenario in mind, two of them are unbalanced with respect to memory usage, violating the size constraints by more than a factor $3\times$. For the other two, our algorithm achieves latency lower by up to 23% (17% on average).

# G  BERT-3 Splits for Throughput Maximization

See Figure 10 for an illustration of an example pair of optimal contiguous (top) and non-contiguous (bottom) splits of an operator-level graph that are returned by our algorithms.

# H  Extensions

In this section, which deals with throughput maximization (i.e. the pipelined setting), we briefly explain how to adjust our model and solutions so as to account for certain different or more complex deployment scenarios that appear in related work or in practice.

## H.1  Interleaving Communication and Computation

Throughout the paper we have assumed that accelerators are invoked when their inputs are ready, at which point they are transferred to the accelerator memory; next, computation takes place; next, outgoing transfers take place (see Section 3). After that, the in-transfer for the subsequent sample/minibatch may begin, and so on. For this reason, the load of a device is defined as the *sum* of the computation cost and the communication cost. However, it is also reasonable to assume that communication (data transfers) may proceed in parallel to computation, at least for different samples. For instance, once we have finished the in-transfer for sample 1, we might simultaneously start the processing of sample 1 and the in-transfer for sample 2. This is the setting considered in the PipeDream paper [HNP+18].

Both of our solutions (DP and IP) can be easily adjusted to this setting: one just needs to define the load of a device as the *maximum* of the computation cost and the communication cost, rather than the sum. In terms of pipeline schedules, one can think of splitting the device into two virtual devices, one holding the communication portion of the load and the other holding the computation portion, that *can* be processing at the same time. Then either virtual device could be a bottleneck in the pipeline.

In fact, one can further assume that the in-transfer and the out-transfer are done over separate channels (full-duplex communication); then a maximum of three quantities (in-transfer cost, computation, out-transfer cost) should be used.

## H.2  Replication

An alternative to model parallelism is data parallelism: an approach where the entire model is replicated over multiple devices that process minibatches in parallel. When using this approach, the communication cost associated with synchronizing the parameters of the entire model proves to be very high for many DNN workloads. Nevertheless, it can also yield large gains for other workloads, especially sparser ones (with a small number of parameters relative to computation cost). PipeDream [HNP+18] proposed a hybrid model-parallel/data-parallel approach, where we form a pipeline, but certain subgraphs in this pipeline can be *replicated* over multiple devices. This allows the automated partitioner to replicate those fragments of the network that will reap the most benefit while keeping synchronization costs low.

We can also introduce this capability into our DP algorithm. When the DP decides whether to place the currently considered subgraph on a CPU or on an accelerator, now it will also decide how many devices to use. That is, in the DP relation, where previously we had $\max\left(\mathrm{dp}[I'][k'-1][\ell'], \mathrm{acc}(I \setminus I')\right)$, now we write[7]

$$\min_{k''=1}^{k'} \max\left(\mathrm{dp}[I'][k'-k''][\ell'], \mathrm{acc}(I \setminus I', k'')\right),$$

where $\mathrm{acc}(I \setminus I', k'')$ is the average time per sample for this subgraph when replicated over $k''$ accelerators. In absence of weight synchronization, this average time would be just $\mathrm{acc}(I \setminus I')/k''$. Weight synchronization (assuming efficient AllReduce collective communication) contributes a term $\left((k''-1) \cdot \sum_{v \in I \setminus I'} m_v\right)/(k'' \cdot B)$, where $m_v$ are sizes of weights associated with nodes and $B$ is the communication bandwidth. Thus, $\mathrm{acc}(I \setminus I', k'')$ should be either the sum or maximum of these two terms, depending on our assumption of interleaving communication with computation (see Section H.1).

This modification of the DP increases the running time by a factor of $O(\max(k, \ell))$. The memory usage remains unchanged.

## H.3  Accelerator Hierarchies

Throughout the paper we have assumed a homogeneous system with $k$ accelerators and $\ell$ CPU cores (probably a single machine). To more precisely capture a distributed setting, one can consider a hierarchical collection of accelerators, such as clusters of GPUs connected internally with faster interconnects and externally (i.e. between clusters) with slower connections (or over a network). Such a multi-level model is used in PipeDream [HNP$^+$18]. Now, the cost of transferring data over an edge between two nodes depends on whether these nodes are placed on devices in the same or different clusters (or even on different machines). The main new challenge is knowing which cost should be taken into account.

The DP solution in PipeDream handles this by dynamically computing optimal splits not only for prefixes of the input network (that correspond to our ideals), but for every contiguous segment of the network. We remark that we can use the same method, at a cost of an $O(\mathcal{I})$-factor increase in both memory usage (number of DP states) and running time.