[Reviews · NeurIPS 2020]

Review 1

Summary and Contributions: Model parallelism is essential for effectively training large models on accelerators with limited memory. Pipelining seems to be one successful way to do this easily. This work posits a Combinatorial Optimization problem which can be solved to find the optimal way to partition a large model across multiple homogeneous devices. == AFTER AUTHOR FEEDBACK == The authors' do a good job of addressing some of the concerns that I have raised in my review. I think my original score still stands. However, I am not fully convinced that all the issues have been sufficiently resolved (cost model vs learning from raw reward signal etc.)

Strengths: [+] This work is very relevant to the current need of model parallelism and pipelining to train large deep neural networks. [+] It proposes a novel Comb. Optimization formulation which can be solved to get the optimal placements. For popular models that get reused thousands of times on standard device configurations, knowing the optimal placement can be very useful. [+] Evaluation is done on some of the recent state-of-the-art models (e.g. BERT). [+] Simple DP / IP formulation means it is easy to implement and try out.

Weaknesses: [-] Number of nodes in the graphs seems to be quite low (~200 for GNMT). GDP reports several tens of thousands of nodes? Is there some manual grouping operation performed on the computational graph? If so, wouldn't this be a significant drawback? Does a "node" correspond to an op run on a device? Few more details on how the graphs are pre-processed could be helpful. [-] Evaluation does not compare with other learning based approaches of model parallelism (for instance to GDP). [-] All the graphs have only up to a few thousand nodes at max, where as the state-of-the-art places graphs with tens of thousands of nodes (see REGAL/GDP/ Hierarchical Device Placer). [-] Unlike the recent line of work using RL. ( see Azalia et al.,), this approach makes use of a fairly sophisticated cost model. How hard is this to build for different configurations? What if we would like to extend this to devices across multiple machines? RL approaches require re-training with the same measured reward signal, where as the proposed approach would need sophisticated config specific modeling.

Correctness: Evaluation methodology is thorough. Claims are easy to verify.

Clarity: Paper is fairly easy to understand.

Relation to Prior Work: This paper does a good job of summarizing and comparing with all the prior work. I do not believe there have been any omissions in this regard.

Reproducibility: Yes

Additional Feedback:


Review 2

Summary and Contributions: This paper proposes an efficient algorithm for model parallelism, which is an interesting topic for model acceleration. They pay attention to two aspects: latency minimization and throughput maximization with the corresponding strategy. The experiments show the efficiency of their proposed methods.

Strengths: 1. Model partitioning across different devices is a practical field and it’s very useful in model deploying for real-world application. Recently, more and more models have huge parameters, which are difficult to be deployed directly on some hardwares. Thus, this problem setting is interesting. 2. They take both contiguous and non-contiguous split scenarios and provide both solution for model deployment. 3. They consider many famous models like BERT and ResNet-50 for experiment. The sufficient results demonstrate the efficiency of their algorithms.

Weaknesses: 1. The algorithms are lack of novelty. Even though the problem setting is a very interesting topic, the proposed algorithms are somewhat trivial. For instance, the algorithm just put different mini-batches on different devices and performs feed-forward and back-forward propagation. 2. I am confused about Figure 3(a). Why the time of sample on Device 2 takes much more time compared the sample on Device 1? And there is no explanation about it.

Correctness: The claims and method are correct and the empirical methodology is correct.

Clarity: The paper is well written.

Relation to Prior Work: It is clearly discussed how this work differs from previous contributions.

Reproducibility: Yes

Additional Feedback:


Review 3

Summary and Contributions: The paper considers the problem of partitioning of a deep neural network (DNN) model on a given set of devices (for e.g. accelerators like GPUs, TPUs etc.) with their memory and interconnect constraints while optimizing certain metrics of interest (for e.g. latency and throughput). Two different settings of non-pipelined model-parallel inference/training and pipelined-parallel inference/training are considered. A combinatorial optimization formulation of the problem along with two algorithms (Dynamic Programming and Integer Programming based) as solutions are provided. Experiments are performed on multiple state-of-the-art DNNs.

Strengths: Soundness of the claims: All the claims are backed by experimental evaluation. A clear description of the Integer programming formulation is provided which is correct. Significance and novelty: Efficient training and inference of DNNs is an important problem. The algorithms are novel for the considered setting. A good feature of the proposed approach is its generality which allows it application to potential any neural-network architecture.

Weaknesses: .

Correctness: Experimental methodology seems correct and the proposed approach shows good improvement over multiple good baselines in terms of both throughput and latency objectives.

Clarity: The paper is well-written with clear description of all the main components of the proposed approach.

Relation to Prior Work: Related work is discussed clearly. Proper description of the differences between the proposed approach and the related work is given. The key distinction is that the proposed method builds a cost model of the metrics of interest and solves an offline optimization problem as compared to previous work that focuses on optimizing the objectives in an online manner while evaluating each design on an actual system or a simulator.

Reproducibility: Yes

Additional Feedback: Update after rebuttal: I didn't have any question and I am happy with author's response. Therefore, my review remains the same.

[Author Response · NeurIPS 2020]

We thank the reviewers for their careful reading and insightful comments. We are encouraged that they all appreciated the relevance, significance, clarity, and demonstrated efficiency of our solution. Reviewers 1 and 4 recognize the novelty of our algorithms; Reviewer 4 points out the generality of our framework. Below we address one general point and some specific questions, but will incorporate all feedback in the final version.

**General comment on scalability.** We evaluated our algorithms on several modern DNN workloads; our algorithms are efficient for all graphs that we tried, which are of size up to 2012 nodes. In addition, after the submission we have discovered a new heuristic that allows one to heavily restrict the search space by linearizing the input graph (essentially adding an auxiliary Hamiltonian path found using a DFS-based topological sort). When this is done, the number of ideals becomes just the number of nodes (plus one), and the Dynamic Programming approach is *guaranteed* to handle graphs of size, say, **50K nodes within minutes**. Strikingly, we have found that with this restriction of the search space, our algorithms *still find the optimal split* for all of the tested workloads! We will add this in the final version.

Looking towards the future and even larger graphs, one can take advantage of the repetitive layer structure of modern (e.g. Transformer-based) models to further shrink the search space.

**Reviewer 1.** *Number of nodes in the graphs seems to be quite low ( 200 for GNMT). GDP reports several tens of thousands of nodes? Is there some manual grouping operation performed on the computational graph? Does a "node" correspond to an op run on a device?* To showcase the flexibility of our approach, we were able to obtain computation graphs of many state-of-the-art models, either on the operator granularity level (where each node corresponds to an operation) or on the layer granularity level (where each node corresponds to a DNN layer / a PyTorch module). Our solvers and cost model are agnostic to the granularity used. Our operator-level graphs are obtained from ONNX Runtime, which does not support GNMT. Instead, we obtain GNMT from PyTorch, which yields coarser layer-level graphs. Note that GDP runs experiments on GNMT-8 (operator graph), whereas we do so for GNMT-**16** (layer graph).

*Few more details on how the graphs are pre-processed could be helpful.* Details about the graph preprocessing are provided in Appendix E.1 (in the supplementary material).

*Evaluation does not compare with other learning based approaches (...)* We compare directly to past work that uses a cost model and employs optimization algorithms to find a split statically before the execution. We also provide a qualitative comparison to the Reinforcement Learning approaches in Section 2. In short, they are online methods that treat the objective function as a black box; while no profiling is needed to build a cost model, this necessitates lots of retraining to evaluate various placements. In contrast, our static technique does not have runtime overhead or require runtime hardware execution changes, which could be very costly on some hardware platforms. Further, our methods find *provably optimal* splits, whereas the RL ones are mostly heuristic, without optimality or approximation guarantees.

*This approach makes use of a fairly sophisticated cost model. How hard is this to build for different configurations?* It is not difficult and in fact quite practical to build an instance of our cost model (see Section 3), as one need only know (for every node) the CPU/accelerator processing time, memory usage, and output activation sizes; for the network we need to know the bandwidth; finally, the number of accelerators and available memory. Most of these parameters can be automatically profiled, measured or estimated. Profiling has been shown to be effective and accurate, as well as practical, for example in PipeDream (which runs profiling and builds a cost model automatically).

*What if we would like to extend this to devices across multiple machines?* Our framework is flexible and readily extends to scenarios exactly like the one you mentioned (hierarchical interconnect topologies); see Appendix H.3.

**Reviewer 2.** *The algorithms are lack of novelty (...) somewhat trivial (...) the algorithm just put different mini-batches on different devices and performs feed-forward and back-forward propagation.* We believe that would be data parallelism; our work in fact addresses the very different notion of [pipelined] *model* parallelism, where the DNN operators are partitioned onto different devices. We would also like to stress that the focus and the contribution of our paper are not the pipelining schemes (due to e.g. GPipe and PipeDream), but novel algorithms that *determine which operators/layers of the DNN should be assigned to which machines*. These algorithms are based on the sound algorithmic principles of Dynamic Programming and Integer Programming and are not trivial. The other reviewers acknowledge their novelty.

The only reason why we discuss pipelining schemes is to argue that our cost model (and the combinatorial optimization problem that we are solving) indeed corresponds precisely to the average time-per-sample that arises when running inference or training using these pipelining schemes. The combinatorial optimization problem that we are solving is also anything but trivial. It is NP-hard for multiple reasons, and its difficulty is further reflected in the performance of multiple baselines that we tested (both simple and more complex ones).

*I am confused about Figure 3(a). Why the time of sample on Device 2 takes much more time compared the sample on Device 1?* The different devices hold disjoint parts of the computation graph, which is partitioned among them. This figure corresponds to an example partitioning with skewed (higher) load on one device (i.e., the time it takes to execute, for one minibatch, those parts of the DNN computation graph that have been placed on that device). We intentionally show a sub-optimal split to highlight that maximum load is indeed the crucial factor that determines the throughput of the system when pipelining is used (being equal to the average time-per-sample – see Sections 5.1–5.3).

[Meta-Review · NeurIPS 2020]

Overall reviews of the paper are positive. However, please do take the reviewers’ feedback into account. In particular, adding more discussion about the scalability of the proposed approach. It would also be useful to have a good discussion about how the proposed approach can be combined with FlexFlow to yield both pipeline and parallelism in other dimensions.